# Brief communication: A magma depletion alternative for vent distribution in volcanic fields

Mark S. Bebbington[1], Melody G. Whitehead[1], Gabor Kereszturi[1]

[1]Volcanic Risk Solutions, School of Agriculture and Environment, Massey University, Palmerston North, 4472, New Zealand

5    *Correspondence to*: Melody G. Whitehead (m.whitehead@massey.ac.nz)

**Abstract**

The location of a volcanic vent controls an eruption's hazards, intensities, and impact. Current kernel density estimation methods of future vent locations in volcanic fields assume that locations with more past-vents are more likely to produce future-vents. We examine an alternative hypothesis that an eruption depletes the magma source, causing holes or dips in the 10    spatial density estimate for future vent locations. This is illustrated with the Auckland Volcanic Field, Aotearoa-New Zealand, where both magmatic and phreatomagmatic eruptions have occurred, according to the vent location, with the latter resulting in more explosive eruptions and hence hazard.

## 1 Introduction

Volcanic fields are regions of distributed volcanism where each vent usually erupts only once, and new eruptions typically 15    occur through new vents in new locations. Approximately 75 volcanic fields have been active in the last 10,000 years with the number of vents in an individual field ranging from tens to thousands (Global Volcanism Program, 2024). Where the next eruption will occur is vital information as the hazards, intensities, and subsequent impact depends on near-surface hydrology, geology and topography, as well as magma composition and eruptive rate (Kereszturi et al. 2014). Furthermore, their typically low average eruption rates ($10^{-4}$ – $10^{-6}$/yr; Valentine and Connor, 2015), and fertile soils, mean that settlements are often 20    located proximal to (or on) these fields. The accuracy of existing approaches to eruption location forecasting, and indeed the fundamental underlying assumptions, remain unknown and unvalidated without any results from prospective forecasts. This technical note outlines a new alternative, magma depletion, modification of existing kernel-based spatial density estimates.

Long-term probabilistic eruption forecasting for volcanic fields is essentially spatial smoothing (Connor and Hill 1995, 25    Marzocchi and Bebbington, 2012; Connor et al. 2015) whereby 2D probability density surfaces are built from the location of known eruption vents. This inherently assumes that locations with more past-vents are more likely to produce future-vents. Kernels are placed coincident with past eruptive centres, and their bandwidths estimated via optimisation algorithms (Connor et al., 2018), e.g., the isotropic Gaussian kernel:

$$\kappa_0(x) = \frac{1}{2\pi h^2} \exp\left[-0.5 \left(\frac{|x_i - x|}{h}\right)^2\right]. \qquad (1)$$

Spatial density at a point is then estimated with
$$\lambda(x) = \frac{1}{N}\sum_{i=1}^{N}\kappa_0(x) = \frac{1}{2\pi h^2 N}\sum_{i=1}^{N} exp\left[-0.5\left(\frac{|x_i - x|}{h}\right)^2\right]. \quad (2)$$

Symbology is consistent throughout this note as follows: $N$: Total number of vents, $h$: isotropic kernel bandwidth, $|x_i - x|$: Distance between $i$ th vent at $x_i$ and location $x$ (in 2D space), $\lambda(x)$: Spatial density estimate at location $x$, $V_i$: Volume erupted at $i$ th vent, $r_i$, $l_i$: radius and height of a volume-equivalent cylinder at $i$ th vent.

## 2 Magma depletion alternative for vent distribution

We examine here an alternative (but not necessarily better) hypothesis of magma depletion, i.e., that after an eruption, the
magma source at depth is depleted in this area, causing holes or dips to appear in the spatial density estimate. These mechanics are aligned to current tectono-magmatic frameworks for volcanic fields (Valentine & Perry, 2006). We assume that the available source region is spatially heterogenous, with past eruption locations representing regions of higher magma fertility (e.g., McGee et al., 2015), but as eruptions occur, they deplete their immediate locality, with larger eruptions depleting the available source region more than smaller ones. This depletion then reduces the likelihood of a further eruption from that area,
although it may still be more likely than some other areas.

Three (of many potential) alternate models are provided here (Fig. 1) that consider different manifestations of how this hypothesis may be pragmatically implemented. These approaches require some changes to the standard mathematical formulae, specifically the calculation of normalisation factors. We also restrict ourselves for this note to variations on isotropic
Gaussian kernels (variations of Eq.1, implemented through Eq. 3) to aid visualisation. The kernels below are used to calculate a spatial density, using a leave-one-out cross validation approach (Bebbington, 2015) that maximises the Kullback-Leibler score
$$S = \sum_k log\hat{\lambda}(x_k) - \int \hat{\lambda}(x)dx, \quad (3)$$
in order to estimate bandwidths and any scaling parameters, where $\hat{\lambda}(x_k) = \sum_{i\neq k}^{N}\kappa_i/Z_i$ (i.e. is computed using all locations except $x_k$), and $Z_i$ is a normalisation factor to ensure the density integrates to 1. Note that the formulae below are independent
of bandwidth estimator (see e.g., Connor et al., 2018 for alternatives to Eq. 3). For the following, $\kappa_0$ is the base kernel (Eq. 1), $\alpha$ is the scaling parameter $[0, \infty]$ (so the simpler case with $\alpha = 0$ is nested in the model), and to collapse the erupted (3D Dense Rock Equivalent) volume ($v_i$) to 2D we set
$$r_i = \sqrt{\frac{v_i}{2\pi l}}, \quad (4)$$
representing a disk of radius $r_i$, where disk height $l$ acts as a scaling parameter $[0, \infty]$ and is optimised as a constant for the volcanic field.

*(1)  Disk-based depletion*
$$\kappa = \kappa_0 \exp\left(-\alpha I(|x_i - x| < r_i)\right), \quad (5)$$
where depletion is applied at all distances $|x_i - x|$ less than $r_i$ (Eq. 4), and with no effect at distances greater than $r_i$, and spatial density at a point is
$$\lambda(x) = \sum_{i=1}^{N}\frac{1}{2\pi h^2 NZ_i} exp\left[-0.5\left(\frac{|x_i-x|}{h}\right)^2\right] * \exp\left(-\alpha I(|x_i - x| < r_i)\right), \quad (6)$$

where $Z_i = \exp\left(-0.5 \frac{r_i^2}{h^2}\right) + \exp(-\alpha)\left(1 - \exp\left(-0.5 \frac{r_i^2}{h^2}\right)\right)$. This partial depletion becomes complete depletion when $\alpha = \infty$.

**(2) Inverse-volume weighted** $\qquad\qquad \kappa = \kappa_0 v_i^{-\alpha},$ (7)

where kernel contributions are down-weighted by volume (larger volume past eruptions contribute less than smaller ones), and

spatial density at a point is $\qquad\qquad \lambda(x) = \sum_{i=1}^{N} \frac{v_i^{-\alpha}}{2\pi h^2 \sum_{k=1}^{N} v_k^{-\alpha}} exp\left[-0.5 \left(\frac{|x_i - x|}{h}\right)^2\right].$ (8)

**(3) Distance-weighted kernel** $\qquad\qquad \kappa = \kappa_0 |x_i - x|,$ (9)

where kernel contributions are down-weighted by volume and multiplied by a distance term $(|x_i - x|)$, with spatial density at

a point calculated by $\qquad\qquad \lambda(x) = \sum_{i=1}^{N} \frac{|x_i - x|}{N\sqrt{2}\,\pi^{3/2} h^3 v_i^{3\alpha}} \exp\left[-0.5 \left(\frac{|x_i - x|}{h v_i^{\alpha}}\right)^2\right].$ (10)

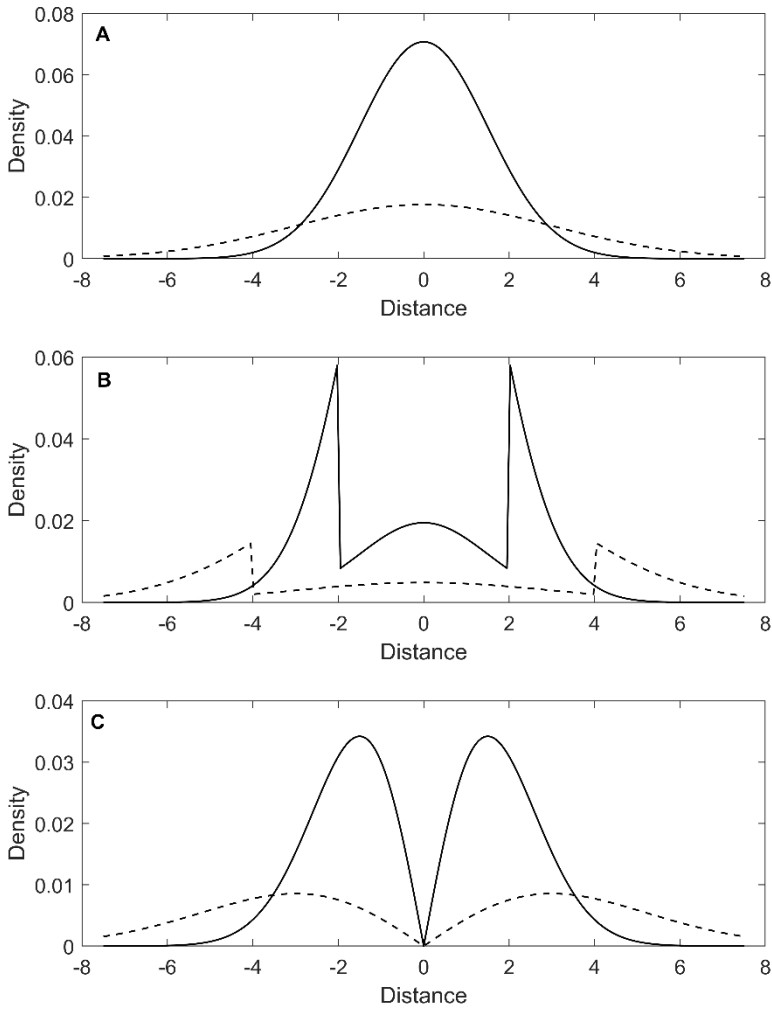


**Figure 1 Cross sections through the centre of the isotropic kernels. A) Isotropic Gaussian (Eq. 1), and Inverse-volume weighted (Eq. 7). B) Disk-based depletion (Eq. 5). C) Distance-multiplied (Eq. 9). The dashed curves represent a bandwidth roughly twice that of the solid curves, noting that for A, the shape of the kernel is identical for Eq. 1 and 7s so only two variations are presented.**

## 3 Example application: Auckland Volcanic Field, Aotearoa-New Zealand

The Auckland Volcanic Field (AVF) has c. 51 eruptive centres of varying ages (~193 ka to ~ 500 yrs BP; Hopkins et al. 2020), and volumes (0.0001 to 0.7 km³, Bebbington, 2015) and lies beneath the City of Auckland (population ~1.7 M). Previous eruptions span the magmatic-phreatomagmatic spectrum with vent locations both on- and off-shore. The most recent eruptive centre was also the largest (Rangitoto, ~500 yrs BP, 0.7 km³), and the most North-Easterly vent. Spatial density estimates for the AVF have been published with both isotropic (Bebbington, 2013) and anisotropic kernels (Bebbington and Cronin, 2011).

For illustrative simplicity, we assume a fixed region of volcanic activity, noting that the delineation of a volcanic field boundary is problematic at best (Bebbington, 2015, Runge et al., 2016). Loglikelihoods for each of the fitted models (Fig. 2) show the fit for all models is similar, with the disk-based depletion model (Eqs. 5 & 6) providing the best-fit to the Auckland Volcanic Field data (Table 1). In terms of the Akaike Information Criterion ($AIC = 2k - 2 * Loglikelihood$, where $k$ is the number of fitted parameters), the additional parameters provide either equal (inverse-volume weighted model) or significant improvement

over the Isotropic Gaussian kernel (Table 1: AIC). The distance multiplied kernel is a poor fit in both relative and absolute terms.

**Table 1: Model loglikelihoods, Akaike Information Criterion (AIC), and fitted parameters**

| Spatial Density Model | Loglikelihood | AIC | Fitted Parameters |
|---|---|---|---|
| Isotropic Gaussian (Eq. 1 & 2) | -322.4 | 646.8 | $h = 2.367$ km |
| Disk-based depletion (Eq. 5 & 6) | -318.6 | 643.2 | $h = 1.538$ km, $\alpha = 6.683$, $l = 0.0053$ km |
| Inverse-volume weighted (Eq. 7 & 8) | -321.4 | 646.8 | $h = 2.395$ km, $\alpha = 0.248$ |
| Distance-multiplied (Eq. 9 & 10) | -324.0 | 652 | $h = 11.65$ km, $\alpha = 0.456$ |

## 4 Discussion

This brief communication introduces a new paradigm, including example formulae and code to produce alternate spatial density estimates for volcanic fields under a magma depletion hypothesis. This shows that the baseline isotropic Gaussian kernel approach can be improved simply by depleting the likelihood in the immediate vicinity of a vent, at the very least, for the AVF. Some of these models can produce estimates with questionable degrees of smoothness (e.g., the ridges in Fig. 2B), future work could introduce additional smoothing parameters – the outstanding problem for this is how to robustly estimate

the appropriate level of smoothing. The formulae trialled here can likely be improved with additional data, for example, the disk-based method could incorporate depletion geometries calculated from the modelling of geochemical data (e.g., major and trace elements).

In the AVF, the widely accepted elliptical boundary (Sporli and Eastwood 1997; Bebbington 2015), provides the justification for finite support, i.e., a fixed subsurface region that reflects the magma source region, from which depletion can then be modelled. This is unusual for volcanic fields (Runge et al. 2014) and is why the AVF was selected as our depletion exemplar. Additionally, vent locations in the AVF appear to be randomly located (homogeneous Poisson distribution – Le Corvec et al., 2013a), rather than clustered or uniformly spaced (Runge et al., 2015), which supports the use of isotropic (symmetric) kernels. This may suggest a uniform, homogeneous source region that is then depleted over time, unlike most volcanic fields that tend to have clustered vent locations (Le Corvec et al., 2013b).

The main assumption in our method is that the process exhibits spatial stationarity. To test this, we refit the four models without Rangitoto (the most recent eruption in the AVF), and got similar ranges and ordering of loglikelihoods, with expected minor changes in fitted parameters (see supplement). Beyond the statistical properties of the model, any underlying hypothesis of spatial density needs to account for stalled eruptions (where magma has left the source region but does not reach the surface). White et al. (2006) suggest that only every $5^{th}$-$10^{th}$ dyke reaches the surface (intrusive:extrusive ratios). Thus, for every observable vent in the depletion model, there may be many others that have depleted the source region but that we cannot see where, or by how much. Hence, we need to assume that stalled eruptions have the same spatial distribution as actual eruptions. In contrast, a clustering model such as kernel density estimation can arise from a homogenous source by assuming that stalled eruptions occur in a different spatial pattern as actual eruptions. The inclusion of eruptive volume as a covariate may improve vent distribution, however, eruptive volumes are often accompanied by large uncertainties (e.g., Kereszturi et al., 2013), especially in older volcanic fields with more burial of the deposits and erosion. However, as long as the relative volumes between vents do not change significantly, then these uncertainties will be absorbed within model fitting parameters, noting that uncertainties in erupted volume estimates will propagate through the model, and consequently may affect uncertainties in the fitted parameters.

These proposed ideas herein have used perturbed kernel-based models (i.e., centred on vent locations), which is obviously a major restriction. The scope for models outside of this class is undelimited, but our results suggest that further reducing the likelihood clustering around previous vent locations might prove fruitful. Future work should also consider a broader family of kernels, including anisotropic, and spatio-temporal options, and the grouping of multi-vent eruptions into single events before spatial density models are deployed (Runge et al., 2014). The effect of such groupings will be most pronounced on those estimation methods that rely heavily on minimum inter-vent distances (e.g., the distance-multiplied model), where likelihoods will be over-penalised by using vents, rather than events. These alternatives may provide insights for those volcanic fields that are purely monogenetic. The appropriate use for caldera systems remains unknown and may prove an interesting avenue for future research, although account would need to be taken of the effect of the caldera structure on vent locations (e.g., Kosik et al., 2020). Volume-based temporal forecasts for polygenetic eruptive centres can be found in Bebbington (2008).

These model outputs are two-dimensional representations of a highly multi-dimensional reality. As such, the inclusion of any available covariates that inform magma source region (e.g., geochemistry – Rowe et al., 2020), system dynamics (e.g., eruptive ages or order – Bebbington & Cronin, 2011), or multi-vent events (e.g., Magill et al., 2005) may be a valuable next step.

**5 Conclusions**

Here, we provide an alternative depletion-based hypothesis for spatial density estimates of vent locations at volcanic fields, alongside the mathematical framework(s), derivations (see supplement) and MATLAB codes required to implement them. Until we can directly observe, monitor and forecast magma source regions in the subsurface, we must investigate as many alternative hypotheses as are plausible, to produce transparent forecasts which together convey accurate descriptions of variability and uncertainty.

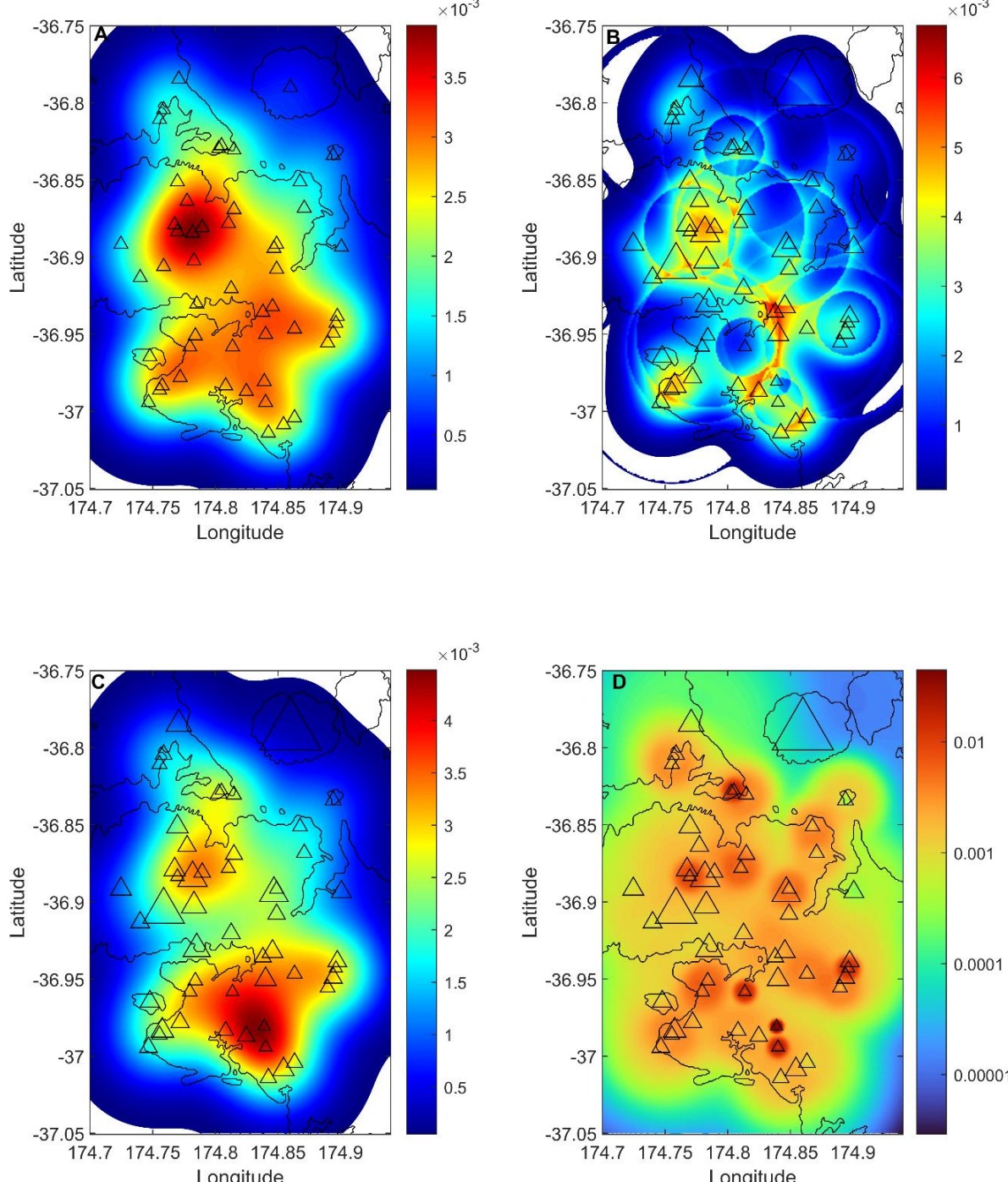


**Figure 2 Fitted spatial densities to the Auckland Volcanic Field, Aotearoa-New Zealand. A) Isotropic Gaussian (Eq. 2), B) Disk-based depletion (Eq. 6), C) Inverse-volume weighted (Eq. 8), D) Distance-multiplied (Eq. 10). Vent location symbols (triangles) scale with square root of volume (except in A as volume is not considered in the model).**

### Data availability

All data (Auckland Volcanic Field eruption centres and volumes) are available in the Supplement.

### Code availability

Code for fitting these equations to the AVF data was written in MATLAB (and can be run in Octave by moving the functions to the top of each script) and is available https://github.com/MelWhitehead/Cheese/.

### Supplement

The supplement related to this article is available online at: << link for nhess supplement to go here >>

### Author contributions

All authors developed and discussed the hypotheses, formulae were derived, checked, (and re-derived), by MB/MW, MATLAB codes were developed by MB, GK provided monogenetic volcanological insight, all authors contributed to writing and editing of the manuscript.

### Competing interests

The authors declare that they have no conflict of interest.

### Acknowledgements

MW is supported by the Ministry of Business Innovation and Employment Smart Ideas Fund (grant number MAUX2301), MW, MB and GK are supported by Ministry of Business Innovation and Employment Endeavour Fund (grant number 160 MAU2444). We thank Nicolas Le Corvec and an anonymous reviewer for their insightful comments that improved this manuscript.

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
