# Peer review of "Brief communication: A magma depletion alternative for vent distribution in volcanic fields"

_EGUsphere, 2025_

## Author Comment (AC1)

We would like to thank both reviewers for their insightful comments and provide responses below. Reviewer comments are in black, author response in green.

**Reviewer 1**

**Review of "Brief communication: A magma depletion alternative for vent distribution in volcanic fields"**

The authors present novel kernel density estimation approaches which adopt volume-dependent annular-shaped kernel functions and inverse-volume weights. The methods are tested on the Auckland Volcanic Field (AVF) vent opening dataset, of which the authors are worldwide experts. Fitting performances of the new methods are compared to a more classical isotropic Gaussian kernel estimation method in terms of log-likelihood and AIC. The authors claim that only one of the three new methods produces a better fitting that an isotropic Gaussian kernel in terms of AIC.

The manuscript is well written, Maths is correct and clear, and the topic is interesting and new. However, results are not yet convincing. Additional testing on other datasets and against other kernel approaches would be advised to complete the study.

MAIN COMMENTS

1. ERUPTED VOLUME UNCERTAINTY

    The authors should acknowledge that in the majority of volcanic fields a great percentage of erupted volumes are uncertain (or unknown). In fact, all the presented methods depend on past erupted volume and therefore depend on the quality of volume estimates, and cannot be applied when the volumes are not available. The effects of the uncertainty on the volume estimates at AVF should be evaluated and, if possible, considered in the likelihood estimates.

MW/MB/GK: This is a good point. What ultimately matters for these methods is whether the *relative* volumes change significantly in response to the uncertainty estimates as this would represent bias. If they do not, then any volume variations / uncertainties will be absorbed by the scalar / fitting parameters in the models. We propose adding the following sentence to the manuscript at line 108:

*"The inclusion of eruptive volume as a covariate may improve vent distribution, however, eruptive volumes are often accompanied by large uncertainties (e.g., Kereszturi et al., 2013), especially in older volcanic fields with more burial of the deposits and erosion. However, as long as the relative volumes between vents do not change significantly, then these uncertainties will be absorbed within model fitting parameters."*

2. ANISOTROPIC GAUSSIAN KERNELS AND OTHER PRE-EXISTING METHODS

    New methods performance should be tested against other well-performing kernel methods, and not only an isotropic Gaussian. Many other approaches have been already tested on the AVF and can be found in references. It would be interesting to compare the volume-depletion family against a broader list of pre-existing methods.

MW/MB/GK: Also a great point, and during the process of writing this brief communication piece we did experiment with anisotropic kernels. The best fitting method in the manuscript was as effective as the standard anisotropic kernel. However, for reasons of both brevity, and due to the apparent randomness of the Auckland Volcanic Field (Le Corvec et al., 2013a), we have left this extension for the next user. We propose the following amendments to lines 95 – 97:

*"Additionally, vent locations in the AVF appear to be randomly located (homogeneous Poisson distribution - Le Corvec et al., 2013a), rather than clustered or uniformly spaced (Runge et al. 2015), which supports the use of isotropic (symmetric) kernels. This may suggest a uniform, homogeneous source region that is then depleted over time, unlike most volcanic fields that tend to have clustered vent distributions (Le Corvec et al., 2013b)."*

As well as the addition of *"a broader family of kernels, including anisotropic, and spatio-temporal options"* to lines 111/112 (future work).

3. VENT CLUSTERS AND POLYGENETIC ERUPTIVE CENTERS

The idea of volume depletion is interesting but conflicts with commonly observed clustering trends and polygenetic centers. Therefore, the method may behave badly for calderas and volcanic fields that are not monogenetic. Testing the methods on other published vent opening dataset could be useful to understand its robustness.

MW/MB/GK: We do not expect this model to be appropriate for polygenetic eruptive centres, nor do we have reason to believe it would work for calderas. We propose adding in the following sentence at Line 114:

*"These alternatives may provide insights for those volcanic fields that are purely monogenetic. The appropriate use for caldera systems remains unknown and may prove an interesting avenue for future research, although account would need to be taken of the effect of the caldera structure on vent locations (e.g., Kosik et al., 2020). Volume-based temporal forecasts for polygenetic eruptive centres can be found in Bebbington (2008)."*

4. DISCONTINUOUS KERNEL FUNCTIONS

The disk-based depletion method is based on kernel functions that are not continuous. Their jumps appear in the convolution results in Figure 2, and create an unrealistic "bubbly" appearance, regardless of its better AIC score. Could you make it smoother without decreasing performance?

MW/MB/GK: A smoothing parameter could be straightforwardly applied to the model outputs, but how this parameter is estimated, and what level of smoothing is geologically-appropriate is a more complex issue, and likely highly subjective. We propose adding in the following addition to the sentence at line 88:

*"Some of these models can provide estimates with questionable degrees of smoothness (e.g., the ridges in Fig. 2B), future work could introduce additional smoothing parameters – the outstanding problem for this is how to robustly estimate the appropriate level of smoothing."*

5. WHY INCLUDING A BAD PERFORMING METHOD

> The distance-multiplied method behaved poorly and I am not convinced of its usefulness in the manuscript.

MW/MB/GK: While we understand the reviewer's perspective, the publishing of methods that do not work can be as important as the publishing of methods that do. This avoids others repeating our mistakes. Additionally, while the distance-multiplied method performs poorly for the AVF, there may be other regions where it does OK.

6. MISSING REFERENCES

> I acknowledge the limitation to 20 references, but I suggest to improve the reference list.

MB/MW/GK: We propose adding the following references (rationale throughout this document):

Bebbington, M.S., Cronin, S.J.: Spatio-temporal hazard estimation in the Auckland Volcanic Field, New Zealand, with a new event-order model, Bull. Volc., 73(55-72), doi: 10.1007/s00445-010-0403-6, 2011.

Bebbington, M.: Incorporating the eruptive history in a stochastic model for volcanic eruptions, J. Volc. Geotherm. Res., 175, 325-333, 2008.Kereszturi, G., Nemeth, K., Cronin, S.J., Agustin-Flores, J., Smith, I.E.M., Lindsay, J.: A model for calculating eruptive volumes for monogenetic volcanoes – Implication for the Quaternary Auckland Volcanic Field, New Zealand, J. Volc. Geotherm. Res., 266(16-33), doi:10.1016/j.jvolgeores.2013.09.003, 2013.

Kósik, S., Bebbington, M. and Németh, K.: Spatio-temporal hazard estimation in the central silicic part of Taupo Volcanic Zone, New Zealand, based on small to medium volume eruptions. Bull. Volc., 82(6), doi:10.1007/s00445-020-01392-6, 2020.

Le Corvec, N., Bebbington, M.S., Lindsay, J.M. and McGee, L.E.: Age, distance, and geochemical evolution within a monogenetic volcanic field: Analyzing patterns in the Auckland Volcanic Field eruption sequence, Geochem. Geophys. Geosys., 14(9), 3648-3665, doi:10.1002/ggge.20223, 2013a.

Magill, C.R., McAneney, K.J., Smith, I.E.M.: Probabilistic assessment of vent locations for the next Auckland volcanic field event, Math. Geol., 37, 227-242

Rowe, M.C., Graham, D.W., Smid, E., McGee, L.: Unusually homogeneous helium isotope composition of the Auckland Volcanic Field and its implications for the underlying mantle, Chem. Geol., 545(119639), doi:10.1016/j.chemgeo.2020.119639, 2020.

7. MISSING COORDINATES

> Please indicate the coordinates and projection of the vent opening data in S3 and clarify their literature source(s) in the caption.

MW/MB/GK: Yes, we will add those.

**Reviewer 2: Nicolas Le Corvec**

This paper provides a valuable conceptual and methodological contribution by proposing a magma depletion-based alternative to standard kernel density estimation (KDE) for vent forecasting in volcanic fields. It is especially commendable for challenging the entrenched assumption that "more vents = higher likelihood," and it offers an implementable framework with code and validation using the Auckland Volcanic Field.

Such contributions are essential for fostering open-mindedness in the scientific community, particularly in probabilistic volcanic hazard modeling, where epistemic uncertainties remain high.

To strengthen the study, it is important to acknowledge and address several key limitations:

1. **Dimensionality Oversimplification**

    The models are inherently two-dimensional, whereas magma ascent and storage processes are fundamentally three-dimensional and evolve over time. This affects how spatial depletion is conceptualized and operationalized. Realistic modeling should account for vertical magma propagation, source region geometry, and the temporal dynamics of recharge and depletion.

MW/MB/GK: We completely agree with the reviewer, and hope to expand our models and research into some number of extra dimensions. The problem very rapidly becomes one of dimensionality, how many parameters need to be fit by an already limited dataset, what parameters are realistic vs what become just a function of each other (e.g., depth vs volume), and the sheer number of unknown unknowns is mind-boggling. However, we propose the inclusion of the following sentence to Line 114 (future work):

*"These model outputs are two-dimensional representations of a highly multi-dimensional reality. As such, the inclusion of any available covariates that inform magma source region (e.g., geochemistry – Rowe et al., 2020), system dynamics (e.g., eruptive ages or order – Bebbington & Cronin, 2011), or multivent events (e.g., Magill et al. 2005) may be a valuable next step"*

2. **Geochemical and Structural Constraints**

    Previous work (e.g., Le Corvec et al., 2013 https://agupubs.onlinelibrary.wiley.com/doi/full/10.1002/ggge.20223) has shown that closely spaced vents in monogenetic fields often share geochemical signatures, suggesting overlapping or connected magma sources rather than isolated zones subject to simple local depletion.

    Additionally, Rangitoto's complex and polygenetic eruptive history (e.g., McGee et al., 2011 https://link.springer.com/article/10.1007/s00410-011-0611-x) challenges the assumption that each eruption entirely depletes a discrete source.

MW/MB/GK: We agree in part but note that none of the models suggest entire depletion, the disk-based model, for example, pushes future events further away from existing events, but

these future events could still feasibly occur at the same location. We agree with the potential addition of external data (e.g., geochemistry) to inform vent groupings, or at the very least, source depths, and think we have incorporated this sufficiently in our proposed additions to Reviewer 2 point 1.

3. **Vent Clustering and Eruption Grouping**

    The treatment of each vent as an independent event may misrepresent eruptive behavior in cases where multi-vent eruptions or spatial clustering occur. Grouping vents into eruption events prior to modeling may be necessary to avoid overestimating spatial depletion.

MW/MB/GK: Please see response to Reviewer 1: 3. VENT CLUSTERS AND POLYGENETIC ERUPTIVE CENTERS. Grouping vents into eruption clusters is itself an uncertain process. Moreover, it then requires a hierarchical model term for the occurrence and location of a vent given an event (see, e.g., Magill et al 2005). This is, similarly to the inclusion of depth (Reviewer 2, point 1) a non-trivial extension, and sufficient data may not exist. Our proposed addition to the text is included above (Reviewer 2, point 1).

**Conclusion**

This work is a welcome methodological advance that opens new avenues for spatial vent forecasting. It is suitable for publication pending minor revisions that acknowledge the dimensionality of the magmatic system and the temporal dynamics of source replenishment.

MW/MB/GK: Thanks 😊